

# A Prequel to the Dantean Anomaly: The Water Seesaw and Droughts of 1302-1307 in Europe

Martin Bauch[1], Thomas Labbé[1,3], Annabell Engel[1], Patric Seifert[2]

[1]Leibniz Institute for the History and Culture of Eastern Europe (GWZO), Leipzig, 04109, Germany
[2]Leibniz Institute for Tropospheric Research (TROPOS), Leipzig, 04318, Germany
[3]Maison des Sciences de l'Homme de Dijon, USR 3516 CNRS, Dijon, 21066, France

*Correspondence to*: Martin Bauch (martin.bauch@uni-leipzig.de)

**Abstract.** The cold/wet anomaly of the 1310s («Dantean anomaly») has attracted a lot of attention from scholars, as it is commonly interpreted as a signal of the transition between the MCA and the LIA. The huge variability that can be observed
during this decade, similarly with the high interannual variability observed in the 1340s, has been highlighted as a side-effect of this rapid climatic transition. In this paper, we demonstrate that a multi-seasonal drought of almost two years occurred in the Mediterranean between 1302 and 1304, and respectively a series of hot and dry summers north of the Alps from 1304 to 1306. We propose to interpret this outstanding dry anomaly, unique in the 13th/14th century, combined with the 1310s and the 1340s cold anomalies, as part of the climatic shift from the MCA to the LIA. Our reconstruction of the predominant weather
patterns of the first decade of the 14th century from documentary and proxy data lead to the identification of multiple European water seesaw events in 1302-1307, with similarities to the seesaw conditions which prevailed in 2018 over continental Europe. It can be debated to which extent the 1302-1307 period can be compared to what is currently discussed regarding the influence of the Arctic amplification phenomenon on the increasing frequency of long-lasting stable weather patterns that occurred since the late 1980s. Additionally, this paper deals with socio-economic and cultural responses to drought risks in the Middle Ages
from contemporary sources and provides evidence that there is a significant correlation between blazes that devastated cities and pronounced dry seasons.

## 1 Introduction & State of the Art

While Medieval studies analyzed since decades the reconstruction (Pfister et al. 1998) and the impact of cold events on premodern societies in the context of the Little Ice Age, few papers focused on droughts, notably from economic history
(Stone 2014). The so-called 'Dantean Anomaly' has been highlighted since the 2000s by Brown (2001) as a wet and fresh anomaly lasting from 1315 to 1321, and leading to famine over NW Europe (Jordan 1998). This climatic anomaly has been recently described more neutrally as 'the 1310s event' (Slavin 2018). A distinctive '1300 event' has been found in proxy data even around the Pacific rim (Nunn 2007). But the focus was always on the cold and wet character of this decade and historians have clearly been fascinated by continuous rains and their impact on food security, often leading to widespread





famines. A lot has been written for example about the deficit of crops and thus of food supply induced by the rainy years
        1315 and 1316, leading to a famine in northern Europe (Campbell 2016; Jordan 1996).

        Yet, in the context of current worries about global warming and its link with a potential multiplication of 2003-like drought
        events, dry periods have found more and more interest from pre-modern climate history (Brázdil et al. 2019; Brázdil et al.
        2018; on the Middle Ages: Rohr et al. 2018). Most research deals with the early modern period though (Garnier 2019;

Munzar 2004; Martin-Vide, Barriendos 1995; Weikinn 1965/66), and the 'millenium drought' of 1540 has been especially
        highlighted (Pfister 2018; Wetter, Pfister 2013; Wetter et al 2014). Concerning the medieval period, the very recent
        publication on the dry year 1473 (Camenisch et al. 2020) is still an exception, preceded only by case studies on medieval
        droughts in Hungary and modern-day Croatia (Kiss 2017; Kiss, Nikolic 2015). In fact, the socio-economic impacts of
        droughts on medieval societies are more difficult to conceptualize than those linked with cold-wet weather. In most parts of

western Europe, droughts were basically benevolent for agricultural production based on cereals and wine (Le Roy Ladurie
        2004), which easily tolerate dry weather conditions if excessive hydric stress doesn't block vegetation growth, as in extreme
        cases like in 1540. Thus, droughts rarely induced socio-economic disasters similar to those frequently associated with wet
        anomalies. As Prybil (2017) states, warm and dry summer half years were for example not dangerous to crops in medieval
        England. They only show an indirect statistical correlation with epidemics, occurring with one year lag, after the introduction

of *yersinia pestis* in 1348, as rodent populations exploded under such conditions. In Mediterranean regions, droughts must
        have been more troublesome for crops and had a direct impact on food-supply and living standards, due to a higher
        vulnerability to lack of water. However, few studies have been devoted to the topic.

        The beginning of the 14th century is considered to be the onset of the transition period from the Medieval Climate Anomaly
        (MCA) towards the Little Ice Age (LIA) and the time around 1300 CE sees the nadir of the Wolf minimum in solar forcing

(Steinhilber et al. 2009). Although these periods are highly disputed and maybe only regionally applicable (Andres, Peltier
        2016; Grove 2001; Pfister et al 1998), this consensus prompted researchers to emphasize mainly the cold and wet conditions
        of the 1310s. It is beyond the scope of our contribution to add to the discussion about the question if there actually was an
        MCA and an LIA and when the LIA started (Bradley, Hughes, Diaz 2003; White 2014). Yet it seems worthwhile to examine
        at least the consistently dry decade (at least during the summers) directly preceding the wet/cold 1310s anomaly. The first

decade of the 14th century actually witnessed two successive major drought events at European scale, one striking Italian
        regions, another impacting regions north of the Alps.

        At the end of his annual report, the anonymous writer of the 'Greater Annals of Colmar', a Dominican monk most interested
        in weather phenomena, stated astonished about the cold season 1303/04: "Winter was cold in Rome, but in Alsace it was
        warm, and on the contrary, it was warm in Rome [the winter before, 1302/03], but cold in Alsace" (Jaffé 1861, 229). It is not

only winter temperature that acted like a North-South seesaw at the beginning of the 14th century all over Europe. As we
        will demonstrate, an even more pronounced water seesaw happened in 1302-1307 with extremely dry conditions in the
        Mediterranean from the end of 1302 to early 1304 while normal humidity levels prevailed north of the Alps, and pronounced
        drought periods in 1304-1307 in most parts of Western and Central Europe. This double event could be by far the longest,



multi-seasonal drought in three European regions within the 13th and 14th century. We therefore want to focus our
reconstruction of this period from documentary and proxy data on Northern and Central Italy (IT), Eastern France (FR) and
Central Europe (CE) (Figure 1). Additionally, there are hardly any studies that focus on the variety of cultural impacts of
drought on medieval societies. Especially regarding the Italian and French material, we can provide first insights.

## 2 Data

### 2.1 Narrative sources

Narrative texts are commonly reckoned in the field of climate history as important sources of information, and historians have
produced for decades almost exhaustive catalogs of events from chronicles and annals. For this paper, we draw principally on
these previous works. Alexandre (1987), Curschmann (1900) and Weikinn (1958, 2017) provide information, mostly with
accurate source criticism, for the territories of nowadays France, Italy, and Central Europe. Additionally, Brázdil and Kotyza
(1995, Appendix I) gathered the material for the Czech Republic, as well as Malewicz (1980) for Poland. Concerning Italy,
where more urban chronicles were produced than elsewhere in Europe, we collected material ourselves, but analyzed also the
unpublished collection of Emanuela Guidoboni (Bologna), covering the period 1000-1500 from about 200 edited narrative
sources, some compilations and thematic articles, and few archival material. We have limited ourselves to the use of critical
editions of contemporary chroniclers out of the Guidoboni collection. In France there are few chronicles, which almost solely
cover the regions of Paris and Alsace during this epoch.

### 2.2 Administrative sources

Of major interest for the history of climate are cities' deliberation protocol books and accounting documentation. As far as
extreme weather events had an impact on the organization of the communities and/or on agrarian production alike, this kind
of documentation provides a bunch of information to the historians of climate. Cities' deliberation protocol books are kept in
Italian archives since the middle of the 13th century. All matters of justice, economy, local policy, social order, etc., were
systematically noted down after each meeting of the town council. Local governments had sometimes to deal with situations
created by climate stress, such as organizing grain trade in case of shortfalls, resolving potential social disorders that came
alongside, organizing processions together with ecclesiastical authorities, or dealing with the disruption of watermills in
cases of floods or droughts. For the city of Siena, we used the unpublished protocols of the *consiglio generale* for the years
1302-1307 (Bowsky 1981).
In France and Germany, such documentation does not exist before the mid-14th century. Nevertheless, rich and unedited
accounting documentation, similar to the one kept in English archives and repeatedly used by historians of climate (Titow
1960; Pribyl 2012 and 2017), can be exhumed for the territory of the County of Savoy. Roll accounts produced by the
administration of the county since the end of the 13th century provide continuous information about the impact of extreme
weather events on local estates' agrarian profits. Wine and cereal productivity, as well as food prices, have been





demonstrated to react to a large extent according to climatic stress in the medieval economy (Pribyl 2017; Camenisch 2015),
and thus yields and price series give at an annual scale resolution an idea of climate trends. Moreover, the accounters who
were responsible for the production of these rolls often referred to climate events to justify any drops of revenue towards the
administration. A close chronology of events can then be drawn from this documentation, which we have been able to
reconstruct for the region of the Bresse (FR).

**2.3 Charters**

The value of charters as sources for climate history has been demonstrated for Hungary (e.g. Kiss 2016, Vadas 2010). While
not revealing any clear evidence for drought, a preliminary search within German charters editions as well as the *Regesta
Imperii* unearthed several accounts in the context of blazes: often indulgences were granted for the reconstruction of buildings
destroyed by the fire. Unfortunately, the charters normally don't give an exact date for the events and therefore only provide
a *terminus ante quem*.

**2.3 Information from manuals**

In addition to information drawn directly from historical sources, for the investigation of blazes and their connection to weather
and climate we made use of existing collections in manual-like literature, as it has been done by Zwierlein (2011) for Early
Modern history, who evaluated the respective sections of the German and Austrian "Städtebücher" (Keyser 1939-1974,
Knittler et al., 1968-2001). For most of the cities included, there is a list of historical blazes – unfortunately often without
information about severity or causes, and always without references. Nonetheless, this huge database is a unique source of
information, especially for the statistical evaluation of blazes and their potential connection to droughts. The "Städtebücher"
are a widely trusted and highly used standard reference for medieval urban history, making its use legitimate.
Additionally, more detailed information about blazes has been drawn from manuals on historical monasteries
("Klosterbücher", e.g. Huschner et al. 2016).

**3 Methodology and results**

**3.1 Reconstruction from narrative sources and creation of climate indices**

We have created, basically according to the well-established methodology (Pfister 1999; Brázdil et al. 2013; Glaser 2013;
general overview: Pfister et al. 2018), climate indices on precipitation and temperature for the period 1290-1320 to show the
dry and hot episodes in the focus of this contribution within their climatical context over three decades. We adapted
established indexing methodology mainly on the temporal scale, as we chose a semi-annual approach ('growing season' vs.
'non-growing season'), according to the focus of documentary records from medieval societies on the agricultural year.
Derived from these well-established climate indices (see SI 1), we closely followed Camenisch, Salvisberg (2020) and





created seasonal drought indices from the above mentioned narrative sources over the longer period 1200-1400 for all three regions (IT, FR, CE) (see SI 1). As the aforementioned sources and climate historical repositories focus on extreme events, we gave values of -3 (extremely dry) and -2 (very dry) if we had actual indicators for agricultural and meteorological drought (Brázdil et al 2019, 75) or a lack of precipitation over 2 months. We have, however, not applied the category of 'socio-economic drought' (Ibid., 75-76), as not all of its indicators are in our opinion specifically related to dry periods. To identify long-term droughts, these drought index values have been accumulated for single years (figs. 8-10); they confirm the

extraordinary character of the 1302-1307 drought events.

If we focus on the precipitation indices only, we get for IT (Fig. 2) a dense picture demonstrating a sustained, almost 24 months long dry period that is not at all represented by the PDSI values from the Old World Drought Atlas (OWDA, Cook et al. 2015). The case of Italian OWDA data is special, as only a handful of dendrochronological series from the Alps and Calabria are available for the period in question. This nourishes previously formulated doubts about the reliability of

simulated precipitation values (Bothe et al 2019), as the OWDA is calculated for large parts of Italy on the basis of only a few tree-ring series.

The precipitation indices for FR (Fig. 3) are scarce, and yet they show a pronounced drought pattern in the growing seasons of 1304-06. In regard to IT, we lack information on the continuity of this drought over the non-growing seasons (1303/04 to 1306/7). But the general tendency is fitting with available OWDA data, except for the summers of 1293 and 1311.

The most interesting results are the precipitation indices for CE (Fig. 4). They have to be stacked, as CE covers a row of quite different sub-regions, yet the tendency is the same in all of those regions. Furthermore, they are mostly consistent with OWDA data points, and if indices seem to differ (e.g. 1291/92, 1294/95, 1305/06), they provide precipitation data on the non-growing season which is not covered by tree-rings.

If we aim for a qualitative description of weather patterns, documentary data provides a clear, reliable and very dense picture

(see SI 2) of meteorological conditions over the period in question. It all started with a rainy summer in 1302 with floods in CE and FR, while the second half of 1302 was already without precipitation in IT. This was followed by a cold winter with freezing rivers (Rhine, Doubs, Adige) and low water levels in CE, FR and IT, while more to the East (Silesia, Russia), the winter was mild and snowless. Spring of 1303 proved cold in CE - and while we have no information on this summer from north of the Alps, IT was hit by continued meteorological, agricultural and hydrological drought the whole year 1303. The

following winter 1303/04 was particularly warm in FR and split in CE: warm in its Western part, cold in Bohemia. A very chilly winter with freezing rivers is reported for IT. Spring and summer 1304 were extremely dry and hot in FR and CE, with all signs of hydrological drought. IT saw strong, yet short precipitation events in late spring, interrupting the 13 months drought, and then again a dry summer until September 1304. Once more, a pronouncedly cold winter 1304/05 followed in FR and CE, strong precipitation in early 1305 in IT that continued into summer and a dry period in summer 1305 in FR. The

winter 1305/06 was so chilly that the Baltic sea froze over and so did rivers in FR and CE and IT. In FR drought continued into spring 1306, and in CE as in IT winter 1306/07 was again very frosty, later changing to flood conditions. In Eastern CE, drought set in in summer 1307, and a heatwave in FR and IT.



### 3.2 Agricultural production in France and England

From the accounting documentation of the Bresse region (Eastern France), we have reconstructed wheat and wine yields for the period 1300-1330. For each of these reconstructions, raw data of different castellanies, i.e. administrative units under the control of a steward, have been extracted and then compiled in aggregate series indexed on the year 1307. Cereal yields have been estimated from the revenue perceived on seignorial lands located in the territories of two castellanies, i.e. Jasseron and Treffort. This information is indirect, as it refers only to taxes perceived on these lands which were cultivated by tenants, and

not to direct indication of cereal quantity harvested each year. Being said, if that induces a limit in the interpretation, the reliability of the reconstructed series is not to dismiss entirely. Wine yield series are much less critical, as the accounters referred directly to the exact amount of wine collected in seignorial vineyards located in four castellanies. Thus, roll accounts allow for a reconstruction of a relevantly detailed chronology of the reaction of the local vineyard toward climate variations. Figure 5 plots the reconstruction of mean wheat and wine yields in the Bresse from 1300 to 1320.

The results show a similar pattern, namely a trend to relatively high yields before 1310 and then a downward trend reflecting the deteriorating weather conditions of the 1310's anomaly. Good harvests, especially for wine, clearly stand out in 1304 and 1305, in response to the successive droughts locally described in Parisian as in Alsatian chronicles (see SI 2). Besides, the plentiful wine harvest of 1304 is confirmed by a contemporary chronicle (Jaffé 1861, 231). In the years 1306/07 vineyard production reaches average values, even if the accounts mention in both years heatwaves in June and/or July. In this case,

though, temperatures reached levels so high that it prevented peasants from properly plowing the vineyard on time, which can explain why production is lower than in previous years. In any case, it can be inferred from the accounting documentation of the Bresse that from 1304 to 1307, summer temperatures certainly reached above-average values. Additionally, the link existing between good wine harvest and warm summer half-year stands out in 1313 too. Accounts mention in that year a summer dryness, which has been benevolent for the vineyard.

Experimental archeology demonstrated the impacts of drought on medieval-style agriculture (Kropp 2019): considerable damage on summer crops, but stable harvests from winter crops. A mixture of different crops guaranteed altogether a sufficient harvest. This can explain the average cereal yields in figure 5, especially in 1304. In contrast, excessive humidity in 1310 and from 1314 to 1316 clearly had a negative impact on the harvest (cereals -20/-40% and wine -80/-60% respectively). Methodologically it is worth noting that cold episodes are mirrored more faithfully through agricultural proxy-

data than dry periods.

We then compared this reconstruction with well-studied and accessible English wheat yields (Campbell 2007). Fig. 6 plots the relationship between cereal productivity in Southern England and the Bresse region (FR), in comparison with East Anglia July-September precipitation indinces reconstructed from local archival sources (Pribyl et al. 2012, Pribyl 2017). From 1300 to 1320 English and French yields correlate significantly (Pearson coefficient r = 0,61). General trends are

similar in the two regions, with average or above-average harvests in the 1300s. Moreover, we find a synchronous movement





between 1304 and 1306 (Fig. 6), reflecting the precipitation trend. The low level of precipitation reconstructed in East-Anglia for these three specific years then most probably applies similarly for the Bresse region, which means that this multi-annual 1304-1306 drought occurred in a large part of NW Europe.

## 3.3 Identification of meteorological patterns

The proxy and documentary data presented in 3.1 and 3.2 provide evidence for the occurrence of an alternating large-scale weather pattern over large parts of Europe between 1300-1310. The found features are similar to the phenomenon of a water seesaw, as it was recently discussed by Toreti et al. (2019) concerning the drought events of 2018 and others of the last 500 years. A water seesaw describes a remarkable dipole of negative water (precipitation) anomalies in one part of Europe and positive ones in another part of Europe. Toreti et al. (2019) associated the 2018 drought to pronounced positive anomalies in

the geopotential height of the 500-hPa level of atmospheric pressure over the continental European landmass north of the Alps. This blocking situation led to the formation of low-pressure anomalies over Northern as well as Southern Europe, with precipitation patterns associated in such a way, that Central Europe suffered a severe lack of precipitation whereas Northern and Southern Europe experienced an excess of precipitation. Thus, in the 2018 case, the water seesaw was positive over Southern Europe and negative over Central Europe.

Similar to what was reported by Toreti et al. (2019), the predominant weather patterns found for the period from 1302-1307 also must have originated from certain seesaw constellations and associated patterns in the geopotential height fields. A possible meteorological interpretation of the reported weather patterns (see SI 2) was found, as is illustrated in Fig. 7a-k and described in the following:

In summer 1302 (Fig. 7a), wet FR and CE and dry IT correspond to a water seesaw which was negative over IT and positive

over FR and CE. Geopotential anomalies were thus positive over IT and negative over FR and CE. Winter 1302/1303 (Fig. 7b) was reported to have been dry and cold over IT, FR and CE, but warm over Western Russia. This situation can be explained by the presence of a blocking large-scale positive anomaly in the 500-hPa geopotential that covered whole IT, FR and Southern CE. A negative anomaly over Northern Europe would have provided such a constellation that warm (and - not reported - likely moist air) would have arrived in Silesia/Russia, as was reported for this season. Thus, a potential seesaw

was tipping from the region CE, IT and FR (negative) towards Eastern Europe, i.e. here Silesia to Russia (positive). Due to the lack of precipitation proxies for Silesia/Russia for the period, this likely constellation can however currently not be proven.

The 1302/1303 winter constellation likely continued throughout the whole of 1303 (Fig. 7c). Continued positive 500-hPa geopotential anomalies over IT, FR, and CE caused a long-lasting cold period in spring north of the alps, while the blockage

led to a continued lack of precipitation over IT. Also in the following winter 1303/1304 (Fig. 7d), the positive 500-hPa geopotential anomaly must have persisted over IT. However, the reported warm conditions in FR and Western CE indicate that the positive anomaly did extend less toward north compared to winter 1302/1303.



Spring 1304 seems to be a turning point for the water seesaw constellation (Fig. 7e). The dryness of FR and CE and the wetness reported for IT let it appear likely that the positive 500-hPa geopotential anomaly moved toward FR and CE (similar

to what was reported by Toreti et al. 2019 for the 2018 drought), enabling precipitation systems to reach IT from the southwest via the Western Mediterranean. The dry summer reported for IT following the wet spring could have been caused by a slight positive 500-hPa geopotential anomaly over IT during this time (Fig. 7f), but in this case, summertime precipitation is rather unlikely anyway in its subtropical climate.

The weather reported for winter 1304/1305 gives a clear indication for the presence of a large-scale positive 500-hPa

geopotential anomaly over Northern Europe and a negative 500-hPa geopotential anomaly over Southern Europe (Fig. 7g). Cold air masses from Eastern Europe were reported for the whole of CE and FR, while IT was reported to be wet. This constellation continued likely during summer 1305, with a slightly increased negative anomaly over eastern CE that allowed normal temperature and precipitation conditions over this region (Fig. 7 h).

Cold air masses from Eastern Europe were reported for the whole of CE, FR and IT in the winter 1305/1306, which must

have been caused by yet another large-scale positive 500-hPa geopotential anomaly over Northern Europe (Fig. 7 i). A negative 500-hPa geopotential anomaly can thus be expected to have been present over Northern Africa. The dry air from Eastern European masses led to the reported dry conditions over FR. That situation likely continued until the winter of 1306/1307, when drought and low temperatures were reported again for CE and IT (Fig. 7 j). The reported increasing flood conditions in IT in spring 1307 can be explained either by melting snow or by a slight movement of the positive 500-hPa

geopotential anomaly towards the north, allowing precipitation systems to reach IT via the Mediterranean.

Finally, the CE drought and heatwave in FR and IT is likely a result of a continuing positive 500-hPa geopotential anomaly over CE (Fig. 7 k). In its center, dryness prevailed while on the western (FR) and southern (IT) borders hot air was advected from Africa.

### 3.4 Correlation of drought periods and city fires

Medieval city fires are a topic touched upon mainly by cultural history (Jenkrift 2003, 83-100; Riegg 2003; Wolf 2015; Wozniak 2011, 2015). But a close connection between drought and fire has been made plausible for extreme years like 1540 (Pfister 2018; Wetter et al. 2014; Mauelshagen 2010, 127-129). Nevertheless, as a general phenomenon this has been put into question (Zwierlein 2011, 102-110), although the latent fire risk of wood-based pre-modern buildings with open fires to heat, cook and provide light is more than obvious (Bitterli 2015; Contessa 2000, 16-18). Already contemporaries saw a close

connection between drought and blaze: "Many cities were consumed by domestic blazes because of the drought and sterility that prevailed in this year" (Wattenbach 1851b, 641).

We have been comparing accumulated drought indices, yet distinguishable by seasons, with the number of blazes we could take from the archives of societies (see sections 2.1, 2.3 and 2.4). A peak in blazes around the 1302-05 drought is visible for IT (Fig. 8), with a significant correlation of droughts and blazes (r = 0,346) over the whole period. The same peak is visible

for FR in 1306 (Fig. 9) with an even higher correlation of fires and droughts (r = 0,657) over the two centuries. The


correlation of drought years and blazes in CE (Fig. 10) is still significant (r = 0,379). Furthermore, we have suspected from these results (Figs. 8-10) that the probability of a blaze might lag by one year the drought event, as wooden structures had dried over long periods and might ignite more easily even with a temporal distance to the drought. We have then tested in a cross-correlation if this assumed connection existed in our data and we found a very significant correlation regarding a one

year lag for FR (r=0,83) and a signifcant one for IT (r=0,59), but none for CE (r=-0,167). The differing results in the case of CE can probably be explained by the non-critical use of documentary data in the 'Deutsche Städtebücher', while relevant blaze information was validated by historical source-criticism for IT and FR.

As the OWDA information for CE (here broadly defined as a rectangle from the Rhine to the Alps to Novgorod in the East and all of Denmark plus Southern Scandinavia to the North, i.e. 47,34° N- 58.69° N, 7.52 ° E - 30.88°E) is relatively dense

and reliable, it has been combined with available information on blazes from documentary data. Yet, the correlation of low PDSI values and blazes in the same years (Fig. 11) is very weak, if not nonexistent (r = -0,06).

What is surprising, is the totally different outcome of the correlation between droughts and blazes for CE, calculated one time from the OWDA-data - with a very bad correlation - and the other time from the drought-indices based on written sources, where the correlation is much better (respectively r = -0,06 and r = 0,379). The data for blazes stayed the same in

both calculations. Here, the discrepancy between the reconstruction from the OWDA on the one hand and the drought indices, on the other hand, comes into play: The OWDA (even more so the maps provided with it) shows a lot more dry periods than documentary data, that mirror mainly outstanding ones. Regarding the misleading picture the OWDA gives for Italy (in comparison with a reconstruction on the basis of written sources, see 3.1.), a general comparison with reconstructions from written sources would be advisable, especially for those regions and periods where there is a good basis

of written sources and where the amount of dendrological data is at the same time rather low (like for Italy).

## 4 Discussion

Finally, we want to highlight aspects of societal impacts and adaptation measures to drought beyond the classical rogation ceremonies and other religious processions (overview Brázdil 2020), especially as we have only one example for them from IT: In May 1303 there were processions for rain in Parma, that indeed 'provoked' a one-day rain (Bonazzi 1902-04, 84). But

there is more to find about how droughts impacted medieval societies, how they were perceived and how contemporaries reacted.

### 4.1 Drought and infrastructural responses

Italy provides a number of infrastructural responses to the drought experience of 1302-04. In 1303, the city of Parma built  a new, larger and deeper fountain (Chron. Parm., 86). Something comparable happened in the Tuscan city of Siena, settled away

from any larger watercourses and traditionally struggling with water scarcity. Not surprisingly, the century-old myth of an underground river below the city, the so-called Diana, was still alive in Siena. In April 1305, one year after the end of the



drought, documentary evidence reveals that the city council actively searched for this underground river by digging in a local church (Bargagli Petrucci 1903, II, 20). An even more real-time response can be found for 1303: In spring, when a dearth of grain already strangled Siena, the city council decided to import grain via the small port of Talamone, 100 km South-West of
Siena (ASS, CG 62, 1303 March 26, c. 99). They had done so already the year before. But in September 1303 - after the successful grain imports that even had created an abundance of food in the city - the city council decided to buy the port for the Republic and to invest heavily in its refurbishment and expansion in the following years (Sordini 2000, 73-112). Sienese citizens were settled in the newly designed city, with infrastructures and military fortifications (Fig. 12). Although Siena had discussed buying the port years before, it was the drought experience and connected food scarcity that let the plans become
real and lead to a long-term infrastructure investment, possibly the most expensive project the Republic of Siena financed outside the capital's walls.

## 4.2 Cultural aspects of drought

The aforementioned infrastructural responses by the city of Siena to the drought experience provoked a satirical response by Dante Alighieri. In his 'Divine comedy'. the Florentine poet mocked his Sienese neighbors with a famous verse: "You will
find them [some Florentines] amongst the foolish crowd [the Sienese] who put their trust in Talamone, and will lose more hope there than in their search for the Diana" (Dante, Divine Comedy, Purgatory, canto XIII). The *Purgatory*, finished during the early 1310s, makes clear that the futile Sienese efforts to search for their underground river did not go unnoticed by their neighbors. Nor did the Florentines ignore the acquisition of Talamone - they had experienced in 1303/04 how vital the port was for their food security, too. So it was just consequent to sign a trade agreement with Siena in August 1311 that
guaranteed Florentine access to maritime grain trade via Talamone, although under conditions very beneficial for Siena (Banchi 1871, 126-127).

In another famous medieval text, the *De regimine principum*, a kind of manual on good governance by Thomas Aquinas and Ptolemy of Lucca, the latter starts writing from 1302 onwards and reflects on the importance of food security: "Food that is sold is not as effective for nourishment as it should be, since it is often adulterated. As Solomon says in Proverbs (Prov. 2,5):
'Drink water from your own cistern', which includes all nourishment, but especially drink, because it can more easily be adulterated. […] There is greater security in using one's own food, since outsiders can easily poison something not kept in its proper storehouse or pantry, and it is more likely to be harmful." (Blythe 1997, 114). The whole reflection about safe access to food and drink is framed with a most striking biblical proverb, traditionally attributed to Solomon himself: "Drink waters out of thine own cistern, and running waters out of thine own well. Let thy fountains be dispersed abroad, and rivers of waters
in the streets." (Prov. 5, 14-16). That Ptolemy of Lucca wrote about the drought in his chronicle (Clavuot 2009, 652) exactly the time he worked on the mirror of princes makes a connection even more plausible.



## 4.3 Societal responses to water scarcity

This worry about access to water, theoretically expressed in the *De regimine principum*, meets social reality in medieval civilization. In times of drought, one of the major problem communities affected had to cope with, in absence of any
efficient water system, was actually entitlement to healthy drinkable resources. In another context, during the extreme drought of 1137, some chroniclers fully described the hardships met by some French rural communities to access water, as people had to walk miles in order to seek for non-dried sources (Labbé 2018). This aspect is complemented by the fact that water sources were frequently polluted with dangerous gastrointestinal pathogens, a situation eventually worsened by high temperatures and low levels of almost stagnant waters in summertime. Thus, in 1307, after three years of drought, a
chronicler of Paris explained that, the vineyard being burnt by frost in April, citizens had to drink water instead of wine, and that an epidemic subsequently broke out during the summer (Buchon 1827, 130). The outbreak of epidemic one year after a warm half summer year, as 1306 may have been, has been emphasized by Pribyl (2017) as one of the most typical consequences of such kind of climatic conditions. The epidemic in Paris in 1307, due to warm conditions during the summer combined with a lack of ersatz drink like wine, can then most probably be interpreted as a result of the back-to-back dry
vintages from 1304 to 1306.

Another side-effect of major droughts in the 14th century was eventually the temporary blockage of production and sometimes even of communication systems. If crops were generally not heavily endangered, at least in the regions north of the Alps, the lack of water induced nevertheless indirectly in some occasions a disruption of the entire food supply system. Difficulties to transform cereals into flour, and to sustain the transportation of food supply from production regions to city
markets could have some economic consequences. This inconvenience was precisely exposed by the Dominican monk of Colmar in 1304: If wheat was quite affordable this year because of benevolent harvests, nevertheless bread was very expensive and scarce at the same time, just because mills settled along dried rivers could not work anymore. Likewise, if winegrowers were able to produce wine of very good quality, as the berries contained a lot of sugar, they could not benefit from it. Actually, wine prices remained very low as shipments were unable to go down the Rhine river and attain the usual
city markets of Strasbourg, Köln, and Trier (Jaffé 1861, 231)

## 4.4 Cultural aspects of blazes in cities

The best documented of all blazes during the 1302-1307 period is the infamous city fire of Florence on 10 June 1304 that burned 1700 houses. It was interpreted by contemporaries as a conspiracy of the political opposition, the Florentine White
Guelfs, which were driven out of the city in the aftermath of the blaze.

The detailed available documentation on 14th-century fires in Florence (Contessa 2000, 89-107) makes clear how unusual a large blaze of this kind was, even for a populous city like the Tuscan metropolis. The first, yet very limited fire prevention policies are known from 1325 (Contessa 2000, 21-27) and consisted of the introduction of brick-built chimneys and stoves. As early as from 1296 onwards, in Siena, the city reimbursed citizens that ruined their tools when fighting fires (Bellissima





1922), and when two blazes hit the city in November 1302, more than 200 people fought the fire in an organized way and used more than 1800 water charges (Agnolo di Tura, 265). These are strong indicators of a proto fire-guard system in Italian city-states around 1300, before official fire guards were established, e.g. 1344-48 in Florence (Contessa 2000, 31-48). One week after the fire, and again in early August 1304, the blaze was even a topic for homilies of the already mentioned Dominican Giordano da Pisa who reminded his fellow citizens that the fire only did what it should do according to God's

will: bring warmth- and when it burned the city's houses, this was no sin, but God's will (Varanini, Baldassari 1993, 314). Another strategy to cope with fire threat was to oblige citizens, in case of droughts or strong winds, to arrange full water buckets at the doors of their houses, ready to be immediately used by anyone in case of fire emergency. A Parisian chronicler described this coping strategy already in 1305, when high temperatures combined with strong wind made authorities worry about a potential disaster (Buchon 1827, 116-117). This is a shared concern of other sources, too (Wattenbach 1851a, 676).

That wind was crucial can also be derived from the fact that the fire was able to cross water bodies like rivers (Wozniak 2015).

**Conclusion**

The wet anomaly of the 1310s has attracted a lot of attention from scholars these last years (Slavin 2019), as it is commonly interpreted as a signal of the transition between the MCA and the LIA (Campbell 2016). The huge variability that can be

observed during this decade, similarly with the high interannual variability observed in the 1340s, have been highlighted as side-effects of this rapid climate change. In the context of global warming, specialists now agree that periods of rapid climate change are accompanied by a probabilistic higher frequency of extreme events (Sippel and Otto 2014). To date, in the field of Medieval climate history, no efforts have been made to underline the outstanding period of drought of the first decade of the 14th century. However, we have demonstrated that two exceptional series of warm and dry summer half-year occurred

respectively in the mediterranean italian regions between 1302 and 1304, and north of the Alps from 1304 to 1306. Following our indices reconstruction from narrative sources, such multi-seasonal and supraregional scale droughts did not occur during the 13th century and do not find any similar case before the 1360-1362 drought that struck all across Europe, more specifically in 1360-61 in Central Europe (Brázdil et al 2020, 82-83; Kiss 2017, 44-45; Bauch 2017, 1102-1104) and England (Pribyl 2017, 102-104), while in 1362 on the Balkan peninsula, around the Black Sea and the Aegean drought

conditions prevailed (Kiss, Nikolic 2015, 13-14). As administrative data gives indicators of drought in Catalonia in 1361/62 (Fynn-Paul 2016, 137), one might even connect a major city fire in Urgell (Battle 1999, 79-82) with this continuous lack of precipitation. There are even indicators for a global dimension of the event: The years 1360 and 1362 were characterized in Japan by major droughts causing famines (Farris 2006, 109), also on the Korean peninsula (Robinson 2009, 163) and also 1362/63 in Western Rajasthan (Rao 2009, 19), which is a considerable difference to the 1302-07 event that is not traceable

outside of Europe.



We propose then to interpret these remarkable conditions, combined with the 1310s and the 1340s cold anomalies, as part of the climatic transition from the MCA to the LIA. More than a last glow of the MCA, the dry anomaly of the 1300s combined with the wet-cold anomaly of the 1310s allow us to speak of a 'long' Dantean Anomaly: Not a new, but now a much more
substantiated starting point for the changing climatic pattern of this period. Our reconstruction of the predominant weather patterns for the first decade of the 14th century from documentary and proxy data likely lead to the identification of multiple European water seesaw events of 1302-1307. The series of reported meteorological conditions for this period show similarities to the seesaw conditions which prevailed in 2018 over continental Europe (Toreli et al., 2019). The period under study was characterized by a series of long-lasting, steady precipitation dipoles which lead regionally to strongly contrasting
precipitation and drought extremes. It can be debated to which extent the 1302-1307 period can be compared to what is currently discussed regarding the influence of the Arctic amplification phenomenon (Cohen et al., 2014) on the increasing frequency of long-lasting stable weather patterns that occurred since the late 1980s. Arctic amplification describes the decrease of the latitudinal temperature gradient between the midlatitudes and the Arctic which was found to weaken the storm tracks, shift the jet streams, and amplify quasi-stationary synoptic-scale atmospheric waves (Coumou et al., 2018).
Future studies should investigate whether such a scenario was also present in the early 14th century when the transition from MCA and LIA occurred. It appears plausible that such a climatological transition is temporally associated with a reduced latitudinal temperature gradient and consequences similar to the currently ongoing Arctic amplification.

Historical sources, coupled with acute source criticism, are deemed to be useful to refine the chronology of extreme events in combination with natural proxy-data. As we have demonstrated, the OWDA tree-ring reconstruction sometimes missed
information due to scarce raw data, especially for southern Europe. An in-depth analysis of narrative and administrative sources, which are sufficiently numerous from the 14th century onward, permits to draw a more accurate picture of this epoch's climate, including winter conditions that must be taken into consideration to get a full image of the droughts' extent. Finally, droughts are phenomena that offer more for a cultural history of climate than just the analysis of religious mitigation strategies. They show both in contemporary perception as in historical analysis a connection to blazes, a major threat to
medieval cities. We could demonstrate for the first time a correlation of droughts and blazes over 200 years, including a one-year lag in these phenomena. Furthermore, drought provokes reflections on thirst and the use of water that we otherwise hardly ever find in medieval texts.

## Author contribution

Martin Bauch provided general conceptualization, curation & analysis of the Italian historical sources, the creation, evaluation,
and visualization of drought indices and city blazes, writing of the original draft and funding acquisition. Thomas Labbé provided conceptual input, curation & analysis of the French historical sources and visualization of other agricultural proxies. Annabell Engel provided curation & analysis of the Central European material and conceptual input on data presentation.



Patric Seifert provided conceptual input on climatological backgrounds, meteorological analysis, and interpretation of documentary data for weather patterns and geopotential maps.


**Acknowledgments**

We thank Andreas Görlitz (Siegen) for help with cross-correlation techniques and advice on statistical issues. Furthermore, we're grateful that we could use the unpublished climate historical collection of Emanuela Guidoboni (Bologna) for the Italian case studies. Finally, we are thankful for input from and discussion with colleagues from the PAGES Working Group CRIAS,

esp. Chantal Camenisch, Rudolf Brázdil and Andrea Kiss during the October 2019 workshop of CRIAS in Leipzig (Germany).

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

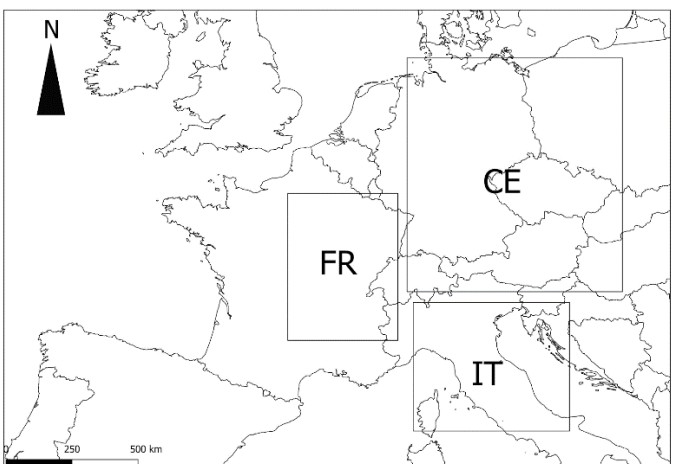

**Figure 1: Geographical zones delimited for documentary researches (Map: Thomas Labbé).**



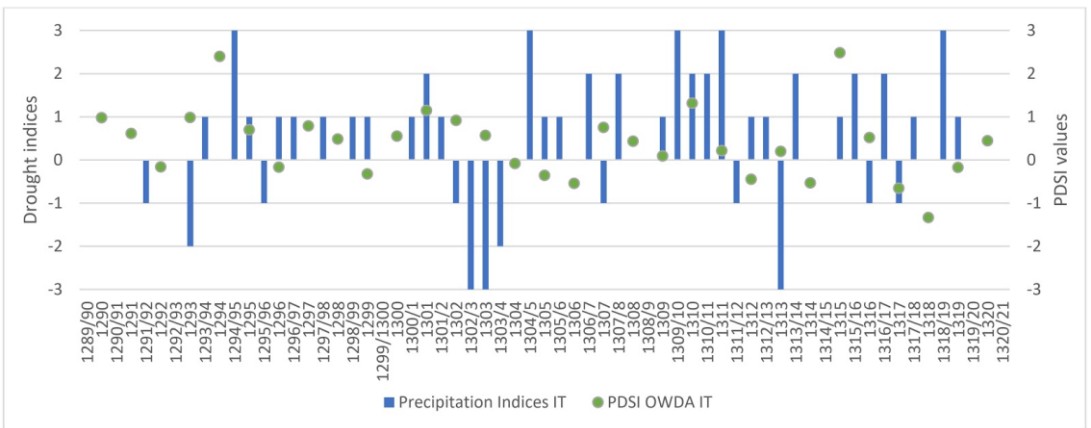

**Figure 2: Reconstructed semi-annual drought indices and OWDA PDSI data for JJA from tree-rings (Cook et al. 2015) for IT, 1290-1320.**

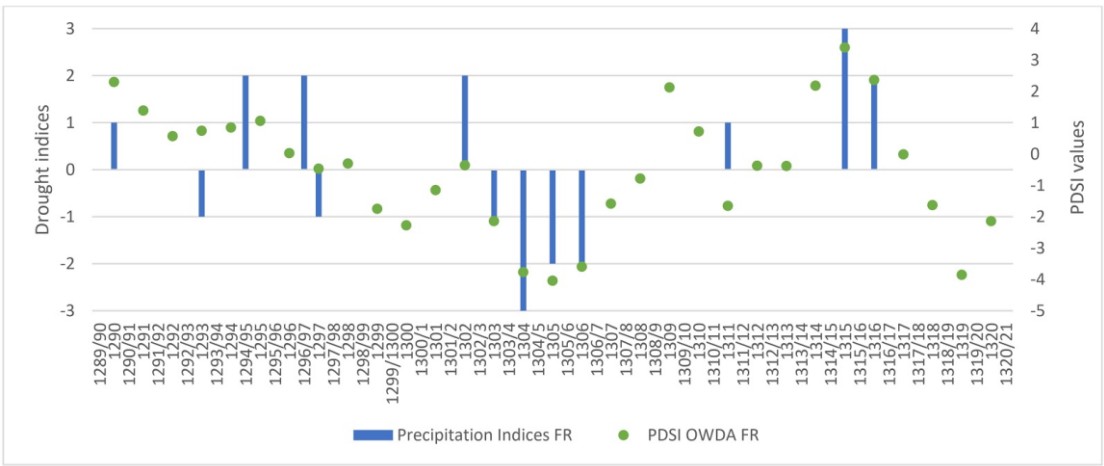

**Figure 3: Reconstructed semi-annual drought indices and OWDA PDSI data for JJA from tree-rings (Cook et al. 2015) for FR, 1290-1320.**




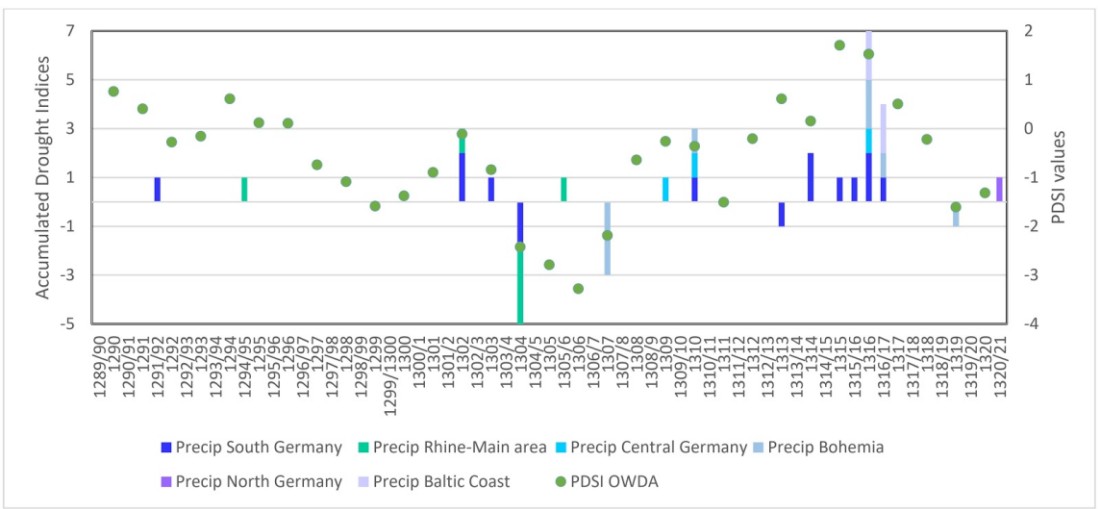


**Figure 4: Reconstructed semi-annual drought indices and OWDA PDSI data for JJA from tree-rings (Cook et al. 2015) for CE, 1290-1320.**

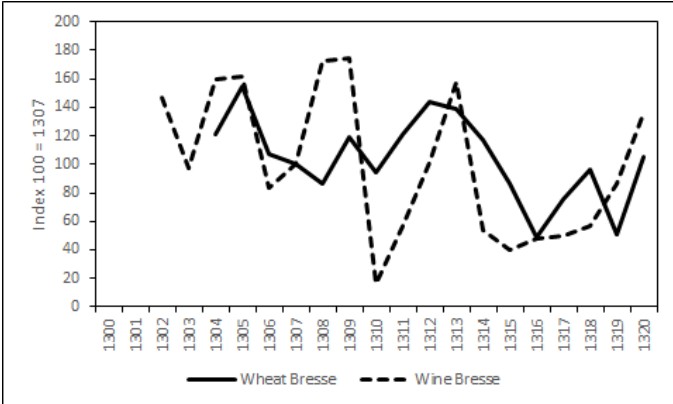

**Figure 5: Mean wheat and wine yield in the region of the Bresse (FR), 1300-1320.**

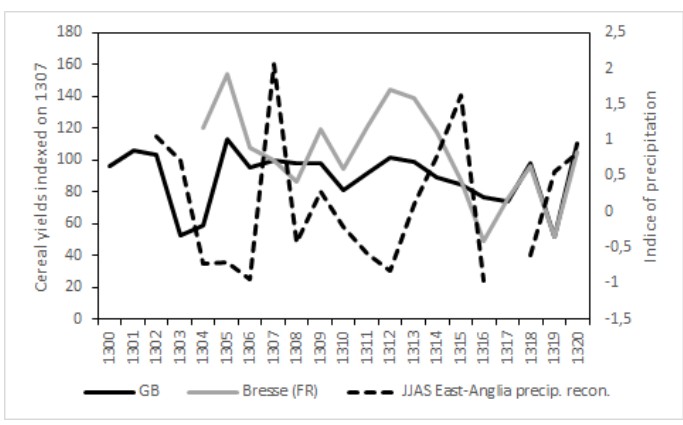


**Figure 6: Comparison of cereal productivity in Southern England and in the Bresse region (FR). Data: Southern England wheat yields (Campbell 2007); Bresse wheat yields (this article); East-Anglia JJAS precipitation reconstruction (Pribyl 2017).**



Figure 7: Illustration of the possible constellations of 500-hPa geopotential anomaly that can explain the weather patterns retrieved from the proxy and documentary sources for the years 1302-1307.



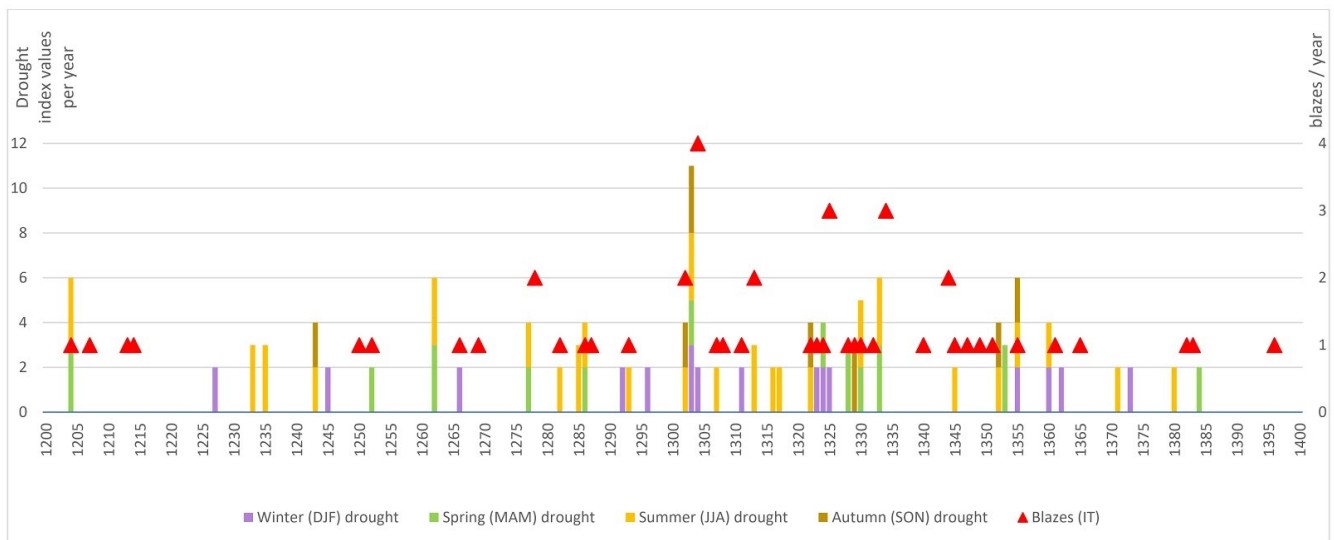

**Figure 8: Reconstruction of drought indices compared with a chronology of blazes for IT, 1200-1400. Data is available for 64 years out of 200 (32% coverage).**


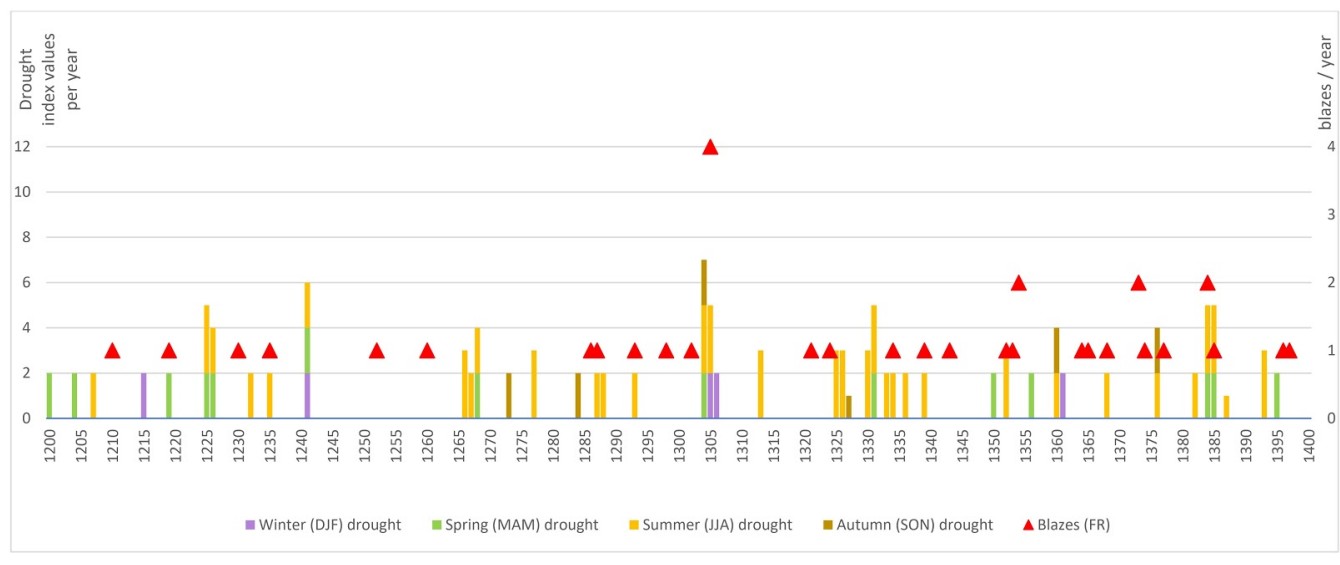

**Figure 9: Reconstruction of drought indices compared with a chronology of blazes for FR, 1200-1400. Data is available for 67 years out of 200 (33,5% coverage).**



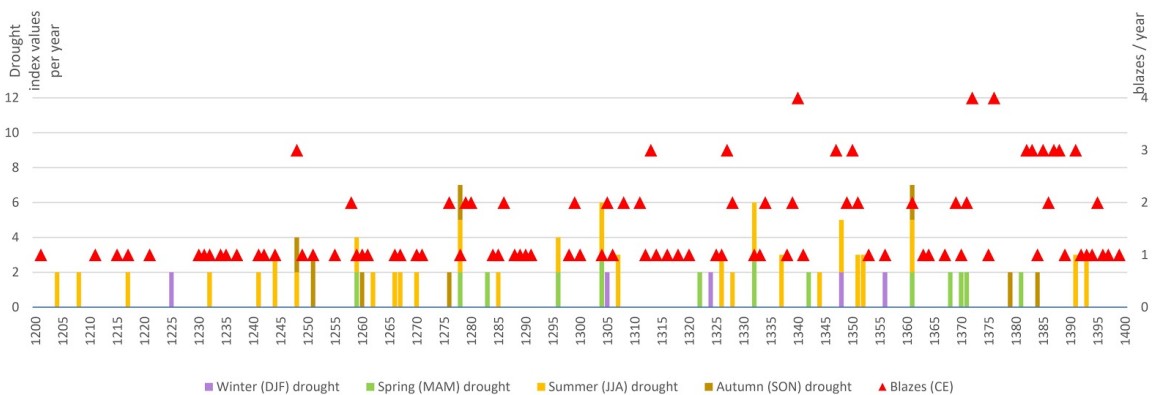

**Figure 10: Reconstruction of drought indices compared with a chronology of blazes for CE, 1200-1400. Data is available for 125 years out of 200 (62,5% coverage).**

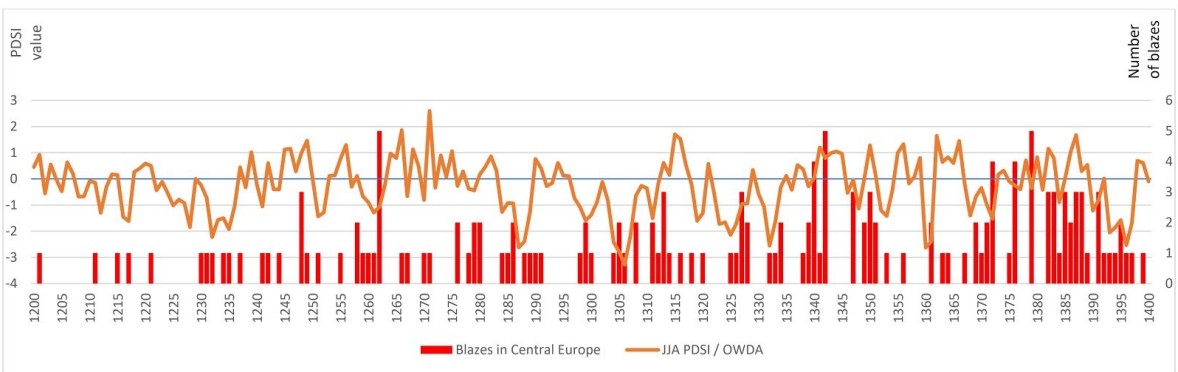

**Figure 11: Reconstruction of JJA precipitation in tree-rings for CE and information on blazes from documentary data for the same region.**

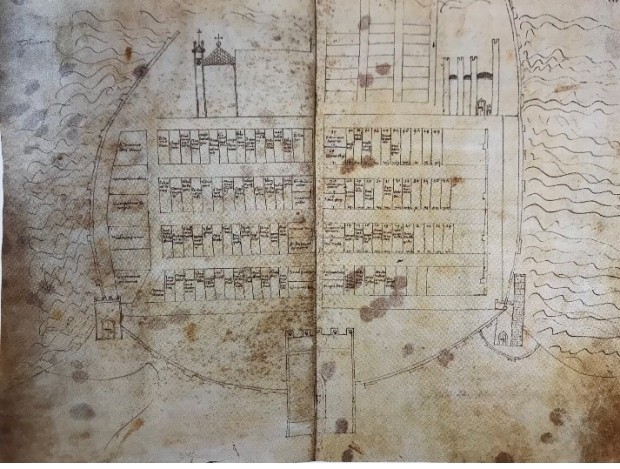

**Figure 12: New city layout of Talamone with plots for new settlers from the mother city. Source: Archivio di Stato di Siena, Caleffo Nero, cap 3, 21 December 1306.**