# Peer review of "A prequel to the Dantean Anomaly: The precipitation seesaw and droughts of 1302 to 1307 in Europe"

_Climate of the Past, 2020_

## Referee Comment (RC1) · Christian Rohr (Referee) · 17 Jun 2020

General comments

This is a very important and innovative exploratory contribution to historical droughts in Europe for a time period not examined in detail so far. The authors testify an excellent overview of the state of the art (see also the large bibliography). The data used for this study is clearly described in chapter 2 and listed in SI 2 (with the original quotes from the narratives sources). Chapter 3 on methodological issues is very accurate and well understandable. It is based on the Pfister temperature and precipitation indices and critically questions the reliability of the Old World Drought Atlas (OWDA), which for the case of Italy is based on only very few tree-ring series. The comparison with the 2018

water seesaw phenomenon (Toreti et al. 2019) is convincing. In the discussion chapter 4, they excellently highlight societal impact issues and different adaptation strategies. The figures are of high quality and well explained in the main text.

Specific comments on individual scientific issues

Abstract:

Please explain the abbreviations MCA and LIA also here for a readership not familiar with this topic so far. This is done only later in the main text (l. 48-49)

Chapter 1: Introduction

l. 55: "major drought events at European scale" seems to be exaggerated to me (and contradictory to the water seesaw phenomenon discussed later) compared to the events of 1360-1362, 1473 and 1540 mentioned in this article. I suggest speaking of "drought events of supra-regional scale" as long as the density of sources cannot be enlarged. Right now, we only know that the in the one case "Italian regions" were affected by drought, and "Regions north of the Alps" in the other case.

Chapter 2 Data

l. 97 The terms "weather" (for short time conditions in the atmosphere) and "climate" (for long-term developments) are not distinguished clearly. I suggest speaking rather of "weather" events" than of "climate events".

l. 100 and 106 chapter 2.3 is doubled. Please rename the latter one into chapter 2.4 (Information from manuals)

l. 101: Concerning the importance of charters I suggest referring also to the monumental monograph by Andrea Kiss published in 2019 ("Floods and Long-Term Water-Level Changes in Medieval Hungary") with a much broader evidence on the rich information from charters.

Chapter 4 Discussion

l. 331-340 It is obvious that very low water had a strong impact on the navigability of rivers and on mills. I wonder whether an increase of ship mills could have been a reaction on the larger rivers such as the Rhine River. Maybe you could check your sources for this detail.

l. 353-354 Giordano da Pisa has not been already mentioned and might refer to an earlier version of the text. Please delete "already mentioned".

Technical corrections

a) Language and style

Both language and style are satisfying. Nevertheless, they should once again be proofread by a professional native speaker. In addition, frequent repetition of the same word should be avoided (e.g. 3 times "reach"/"reached" in l. 174-177). In some cases, "climate" used as adjective should be replaced by "climatic". "impact" should only be used in singular (l. 38, 66, 180, 277).

b) Formal requirements

Revision is needed, however, concerning the formal requirements. Numerous references in the text and in the bibliography are not accurate enough and show a relatively large number of typos:

l. 23 and 51 "Pfister et al. 1998" cited twice cannot be found in the bibliography, whereas in the bibliography (l. 539-540) Pfister, Schwarz-Zanetti, Wegmann 1996 can be found, which is nowhere else cited. I suppose "Pfister et al. 1998 refers to this title, but the authors have to clarify it.

l. 26, 31 and 511-512: Jordan's book has been published in 1996, but in l. 26 it is cited as "Jordan 1998" and in the full bibliographical entry as well. I suppose that the authors refer only to one title published in 1996, but they should clarify and correct the typos.

l. 53 and 596: Sam White's article "The real Little Ice Age" in quoted as "White 2014

in l. 53, but as "White 2013" in the full bibliographical entry. The case is a bit tricky, because the volume of the "Journal of Interdisciplinary History" has been published in 2014 (see also the front matter), but in the baseline of the first page of the article, the misleading information "© 2013" is given. I suggest unifying to "2014".

l. 278: "Brázdil 2020" is missing in the bibliography

l. 284, 290 and 301: Sources are given by a short quote, but a full bibliographical reference is missing. In the first case "Chron. Parm., 86" should be replaced by "Bonazzi 1902, 86"

l. 373 "Brázdil et al. 2020, 82-83" is not cited in the bibliography. Do you mean "Brázdil et al. 2019"? Please clarify.

l. 477-478 and 488-489: Normally all authors are named in the full bibliographical entry. Please unify also for the further authors of these two publications and replace "et al.".

Major typos and formal inaccuracies to be corrected:

l. 35 read "Martín-Vide, Barriendos Vallvé 1995" instead of "Martin-Vide, Barriendos 1995"

l. 35 and 538 read "millennium" instead of "millennium"

l. 38 and 375 read "Nicolić" instead of "Nicolic"

l. 43 read "Pribyl" instead of "Prybil"

l. 92 and 188 read "Pribyl et al. 2012" instead of "Pribyl 2012"

l. 188 read "indices" instead of "indinces"

l. 196 read "patterns" instead of "pattern"

l. 245 read "Jankrift" instead of "Jenkrift"

l. 279 read "Bonazzi 1902, 84" instead of "Bonazzi 1902-04, 84"

l. 348 read "Ibid., 21-27" instead of "Contessa 2000, 21-27"

l. 351 read "Di Tura del Grasso 1839, 265" instead of "Agnolo di Tura, 265"

l. 370 read "Italian" instead of "italian"

l. 440-441 read "Michael Imhof Verlag" instead of "Dr. Imhof" (cf. l. 599 and 603)

l. 448 read "Bothe, W. M." instead of "Bothe, William Marvin"

l. 458 read "Barriendos, M." instead of "Barriendos"

l. 465 add series title after "Jahrhundert": "WSU. Wirtschafts-, Sozial- und Umwelt-geschichte 5"

l. 478 and 489: set year after comma, but not in brackets

l. 513 delete "(1939-1974)"

l. 534 read "Martín-Vide, J. and Barriendos Vallvé, M." instead of "Martn-Vide, J. and Vallv, M. B."

l. 544 delete "365"

l. 561 read "Sippel, S. and Otto, F. E. L." instead of "Sippel, S., Otto, F.E.L."

l. 576 read "Baldassari" instead of "Baldassarri"

l. 579 read "1851a" instead of "1851"

l. 581 read "1851b" instead of "1851"

l. 593 read "Nordli" instead of "Nordl"

l. 599 and 603 read "Wagener" instead of "Wagner"

Some general remarks concerning unification of style:

a) Hyphens for page and year numbers are sometimes short, sometimes long. Please

unify.

b) Short quotes are partly divided by semicolons and partly by commas. Please unify.

c) "et al." is mostly written with a full stop after "al", but sometimes not. Please unify.

d) "edited by" is sometimes written with a colon at the end, sometimes not. Please unify.

Overall assessment

In general, this is an important and innovative contribution, which should be accepted with minor revision, i.e. there are mostly some technical improvements (formal requirements, typos) and some clarifications needed, as mentioned in my comments.

---

## Referee Comment (RC2) · Anonymous Referee #2 · 24 Jun 2020

This paper deals with a climate reconstruction for the period from 1290 to 1320 CE, focusing on droughts and dry years in this period. In addition, societal climate impacts in these years were discussed.

It is undoubtedly a very interesting paper with remarkable results. The methods are convincing and appropriate, the graphs are of good quality. I am sure that this paper is in many ways an important contribution to climate history of the late Middle Ages.

However, the paper has a rather unusual structure for a journal like Climate of the Past. Therefore, some changes would be highly recommended in this respect. The relevant information is actually all provided in the text, but it should be structured in a slightly different way. This should be possible with a minor revision.

[Figure]

The introduction should better clarify the key question and objectives of the paper. Elements of this can now be found in several places. A short outline at the end of this introduction might also be useful.

In chapter 3, methods and results should be separated, as these chapters have a clear and distinct function which should not be mixed up. Elements of the methods are to be found throughout the chapter and would be more clearly presented and appreciated in a separate chapter before the results.

Comparisons with other existing series, such as the OWDA or Campbell 2007, could, however, be moved to the discussion section.

Most of the sections with climate responses, which is now in the discussion chapter, is actually part of the results, as far as I understand. If this is the case, it should be moved to the respective chapter.

It should also be made clearer where the long series of blazes for the statistical comparison come from. As I understand, this information is derived from the same sources as the other data (lines 253-254). If so, it would be useful to briefly describe in the methods section how this data was collected. If the majority of the data comes from the literature, this should be better indicated.

In the conclusion, comparisons with Japan, the Balkans, Catalonia, etc. appear for the first time. It would be better to present them in the discussion chapter first.

Perhaps, it might be worth considering whether the state of the art should refer to the work on droughts in Spain by Barriendos and others.

However, I congratulate the authors on these very interesting and important results and recommend that the structure of the paper be revised again with a view to clarity and logical sequence.

---

## Author Response (AR1)

GWZO, Reichsstraße 4–6, 04109 Leipzig

Prof. Stefan Grab
University of Whitswatersrand
Johannesburg, South Africa

Stefan.grab@wits.ac.za

Leipzig, 15.09.2020
Betreff:
Author's response

Dear Stefan,

please find attached our responses to the Referee comments and a marked-up version of our document. We haven't changed the figures, but apart from that, large parts of the manuscript structure have been revised according to RC2 and our own critical review of our first manuscript. And of course we integrated the feedback of RC1 (Christian Rohr), which required only minor formal changes. Furthermore, we had a thorough language copy-editing by a native speaker with a professional background in academic research in history. As linguistic quality was a key critique to our paper, well deserved, we hope to have changed it and present an article in much better quality now.

Supplementary Information is attached as a zip file of one PDF and one Excel file.

Finally, we present our submission as a zipped LaTeX-package, including the images (as jpgs and pngs) and their captions in a separate folder (/images). If resolution does not suffice, please let us know and we'll see that we can do

The revised article can be found named as template.pdf or template.tex respectively in the main folder. To provide a possibility to countercheck, we attached the PDF of our revised Word-File including the images in the zip file of the LaTeX version:
Bauch Labbé Engel Seifert – A prequel to the Dantean anomal.pdf

I hope that's sufficient. Please come back to me if any questions are still open.

Thanks for all your efforts with our article,

Leibniz-Institut für
Geschichte und Kultur
des östlichen Europa
(GWZO) e.V.

Specks Hof (Eingang A)
Reichsstraße 4–6, 04109 Leipzig
Tel. +49 341 9735 587
Fax +49 341 9735 569
info@leibniz-gwzo.de
www.leibniz-gwzo.de

**Dr. Martin Bauch**
Projektleiter
Nachwuchsforschungsgruppe
"The Dantean Anomaly 1309-1321"
martin.bauch@leibniz-gwzo.de

Vereinsregister: Amtsgericht Leipzig, Registernummer: VR 2617
Finanzamt Leipzig II, St. Nr. 231/140/27308, Bankverbindung: Sparkasse Leipzig,
BIC: WELADE8LXXX, IBAN: DE21 8605 5592 1100 2775 24

Clim. Past Discuss.,
https://doi.org/10.5194/cp-2020-34-AC1, 2020

[Figure]

Thank you very much for this constructive and informative review. We could easily agree upon all your revision proposals with regard to terminology (weather vs. climate, MCA, LIA) and the research literature we could still add (Kiss 2019).

We agreed that it is better to describe the drought event as "supra-regional event, maybe even of transcontinental scale", as data ertainly covers mainly Central Europe, France and Italy, with some indications from England, but - and this we added newly - also from the Middle East: 1304-06 CE are drought years in Syria and Egypt, with roga-tion processions for rain from Damascus (see on this Vogt et al. 2016, p. 91; Raphael 2013, 96-96)) and low-water levels of the Nile (Chalyan-Daffner 2013, pp. 565, 668;

[Figure]

Vogt et al. 2016, p. 91). While documentary data for the Byzantine area in between Italy and the Middle East remains silent on drought between 1302-07 (Telelis 2004, vol. 2, No. 626-627), proxy data from the Aegean can help: An annual precipitation reconstruction from North Aegean tree rings (Griggs et al. 2007) demonstrates that the years 1302-04 are among the five driest periods in the 13th and 14th century in this region.

Raphael, Sarah Kate (2013): Climate and Political Climate. Environmental Disasters in the Medieval Levant. Leiden, Boston: Brill.

Chalyan-Daffner, Kristine (2013): Natural disasters in Mamlǻńk Egypt (1250 - 1517). perceptions, interpretations and human responses. Ruprecht-Karls-Universität, Heidelberg.

Vogt, Steffen; Glaser, Rüdiger; Kahle, Michael; Hologa, Rafael; Münch, L.; Erfurt, M. et al. (2016): The Grotzfeld Data Set - Coded Environmental, Climatological and Societal data for the Near and Middle East from AD 801 to 1821. In: Rüdiger Glaser, Michael Kahle und Rafael Hologa (Hg.): tambora.org data series. vol. I. Online verfügbar unter doi:10.6094/tambora.org/2016/c156/serie.pdf.

Telelis, Ioannis G. (2004): Îl J$_{\varepsilon\tau\varepsilon\omega}$ΣλΣ$_{\gamma\iota\kappa}$Îř $\varphi\alpha\iota\nu$ÏŇ$_{\mu\varepsilon\nu\alpha}$ $\kappa\alpha\iota$ $\kappa\lambda$Îŕ$_{\mu\alpha}$ $\sigma\tau$Σ Î$\check{S}_{\upsilon\zeta}$Îř$_{\nu\tau\iota}$Σ. 2 vols, Athens: Akademia Athinon.

Griggs, Carol; DeGaetano, Arthur; Kuniholm, Peter; Newton, Maryanne (2007): A regional high-frequency reconstruction of May-June precipitation in the north Aegean from oak tree rings,A.D. 1089-1989. In: Int. J. Climatol. 27 (8), S. 1075-1089. DOI: 10.1002/joc.1459.

With this further information and bibliography added, we'd like to adapt our formulation and speak of a supra-regional, maybe trans-continental drought event in 1302-04.

We couldn't find evidence if low water levels led to an increased use of ship mills, but the hypothesis seems plausible and wort further research in the future, but probably on

periods with a denser record on mill construction.

Furthermore, we fully agree with the more technical corrections on language, style and formal requirements. We'll change the text accordingly and have it proof-read by a professional native speaker.

———————————————

[Figure]

Clim. Past Discuss.,
https://doi.org/10.5194/cp-2020-34-AC2, 2020

[Figure]

**CPD**

we discussed your valuable review among us authors and agreed that we can integrate almost all of your proposals to improve our text. Thank you very much for your effort and the important input.

So we will clarify key questions and provide a better outline in the introduction. We'll split up chapter 3 in methods and a new chapter 4 (results) and move the already present text blocks accordingly. Still, we would like to keep the comparisons of our results with OWDA and Campbell 2007 in the results section, as the accordance of our results and those of other colleagues is a key finding of our manuscript. Yet, we will add

a paragraph on chapter 5 (discussions) arguing for a systematic counter-check of (at times relativcely scarce) dendrochronological reconstructions with dense documentary data (whereever possible). As proposed, we'll move societal drought responses mostly to chapter 4.

We were also convinced that is is usefula to add more bibliographical information on where from we took the data for the long series of blazes from and how we collected it (chapter 3).

And we found it very reasonable to put our comparative section on the 1361-62 event with examples from within and beyond Europe in the Discussion chapter 5.

With regard to the work of Manuel Barriendos (Vallvé), he is main or co-author of three titles in our bibliography and in the state of the art section, as he's a main contributor to Brázdil et al 2019, which provides the best overview on his important work on droughts, but also on the general history of drought. As rogation ceremonies are not at the core of our paper, we hope it is justified not to quote further of his certainly key publications in this part of historiography of droughts.

We hope that the paper will gain by these modifications a better, more logical structure, more adapted to publication habits in the sciences and particularly in Climate of the Past.

[revised manuscript text omitted]

65 Schwarz-Zanetti, Wegmann 1996), this consensus has prompted researchers to emphasize mainly the cold and, wet conditions of the 1310s. It is beyond the scope of our contributionThis article does not intend to add to the discussion about the question if there actually was an over whether the MCA and an LIA andactually occurred and, if so, when the LIA started (Bradley, Hughes, Diaz 2003; White 2014). Yet), but it seems worthwhile to examine does seem worth examining at least the consistently dry decade (at least during the summers) which directly preceeedingpreceded the wet, cold period of the

70 1310s anomaly. As . In fact, in the first decade of the 14thfourteenth century actually witnessed, two successive major drought events of at least supra-regional scale affected Europe, one striking Italian regions, another impacting regions north of the Alps, we aim with this. This article at reconstructing itsaims both to reconstruct their duration, extensionextent, and severity, look for and to examine the related socio-economicsocioeconomic impacts and socio-cultural reactions. Furthermore, we provide an estimate of the temporal evolutionIt also provides an an approximate timeline of the underlying

75 meteorological patterns underlying the observed anomaliesand putand contextualizes these into context withanomalies by comparing them to similar events reported for the time range between 1200– and 1400 CE.

At

In 1304, in the endconclusion of his annual report, the anonymous writerauthor of the 'Greater Annals of Colmar'Colmar, a Dominican monk most interested in weather phenomena, stated astonished about the cold seasonwith

80 astonishment: "[This past] winter [i.e. 1303/04: "Winter] was cold in Rome, but in Alsace it was warm, and on the contrary, [the year before, i.e., 1302/03] it was warm in Rome [the winter before, 1302/03],, but cold in Alsace" (Jaffé 1861, 229). It is not only winter temperaturetemperatures that acted like a North-Southnorth-south seesaw at the beginning of the 14thfourteenth century all over Europe. As we will demonstrate, there was an even more pronounced waterprecipitation seesaw happened infrom 1302– to 1307, with extremely dry conditions in the Mediterranean from the end of 1302 to early

85 1304 while normal humidity levels prevailed north of the Alps, and pronounced drought periods infrom 1304- to 1307 in most parts of Westernwestern and Centralcentral Europe. We will examine if this double event couldmight have been by far the longest, multi-seasonal drought by far in three European regions withinduring the 13ththirteenth and 14th century. We therefore want to focus ourfourteenth centuries. Our reconstruction of this period fromis based on documentary and proxy data on Northernnorthern and Centralcentral Italy (IT), Easterneastern France (FR) and Centralcentral Europe (CE) (Fig. 1).

90 Additionally, there are hardly anyThere is a dearth of studies that focus on the variety of cultural impacts of drought on medieval societies. Especially regarding, but particular in regard to the Italian and French material, we can provide firstsome initial insights into the economic impact,impacts and stress these drought-stress imposed on medieval societies and adaptivehow they adapted with measures like fire-fightingfirefighting and infrastructuresinfrastructure to improve access to water and food, cultural implications of drought.

95 The article is structured as follow: In Section 2 we provide a descriptiondescribes and critical evaluation ofevaluates the used 
[revised manuscript text omitted]
 ITdropped unusually low, while more to the Eastareas further east (Silesia, Russia), the winter was ) experienced a mild and snowless. Spring winter. The spring of 1303 proved cold in CE—and while we have no information on this summer from north of the Alps, IT was hit by continued endured a meteorological, agricultural, and hydrological drought the

265  whole year for all of 1303. The following winter (1303/04) was particularly warm in FR and split, while conditions in CE: varied from warm in its Western part, western regions to cold in Bohemia. AMeanwhile, IT is reported to have had a very chilly winter with freezing rivers is reported for IT.. Spring and summer 1304 were extremely dry and hot in FR and CE, with all signs of hydrological drought. IT saw strongsignificant, yet short, precipitation events in late spring, interrupting the 13 months of drought, and then again aanother dry summer until September 1304. Once more, a pronouncedly cold winter in

270  1304/05 followed in FR and CE, with strong precipitation in IT in early 1305 in IT that continued into summer and a, while FR faced another dry period in summer 1305 in FR. The winter of 1305/06 was so chilly that the Baltic seaSea froze over and so, as did rivers in FR and, CE, and IT. In FR, drought continued into spring 1306, and, in both CE as inand IT, the winter of 1306/07 was again very frosty,cold, followed later changing to flood conditions.by flooding. In Easterneastern CE, drought set in in summer 1307, andas a heatwave inswept across FR and IT.

275  If we leave asidelook beyond the core regions of this study and look as far as to the Middle East, we see that 1304 06 CE are drought years in Syria and Egypt also experienced drought conditions in 1304–1306 CE, with rogation processions for rain fromin Damascus (Vogt et al. 2016, 91; Raphael 2013, 96-96) and low water levels ofalong the Nile (Chalyan-Daffner 2013, 565, 668; Vogt et al. 2016, 91). While documentary data for the Byzantine area in regions between Italy and the Middle East remains silent ondoes not mention drought between 1302 07 and 1307 (Telelis 2004, vol. 2, No. 626 627),

280  proxy data from the Aegean basin can help: Ana reconstruction of annual precipitation reconstruction from based on North Aegean tree rings (Griggs et al. 2007) demonstratessuggests that the years 1302 04 –1304 are among the five driest periods inof the 13ththirteenth and 14thfourteenth century in this region.

Hence we We might thus describe the drought event not only as supra-regional, but maybe even perhaps as an phenomenophenomenon of transcontinental scale.

285

**4.2 Agricultural productionProduction in France and England**

Figure 5 plots the reconstruction of mean wheat and wine yields in the Bresse from 1300 to 1320. The resultsBoth lines show a similar pattern, namely a trend to relatively high yields before 1310 and then a downward trend reflecting the deteriorating weather conditions of the 1310's1310s' anomaly. Good harvests, especially for wine, clearly stand out in 1304

290  and 1305, in response to the successive droughts locally described in both Parisian as inand Alsatian chronicles (see SI 2). Besides, the plentiful wine harvest of 1304 is confirmed byIn addition, a contemporary chronicle (Jaffé 1861, 231).) confirms the plentiful wine harvest of 1304. In the years 1306/07 vineyard and 1307, vineyards' production metwas fairly average values, even if thethough historical accounts mention in both years heatwaves in June and/or July. In this case,

though of both years—i.e., generally favorable conditions for vineyards. In these years, however, temperatures were so high

295  that it prevented peasants from properly plowingcould not plow the vineyard on time, which canmight explain why

production iswas lower than inthe previous years. In any case, it can be inferred from the accounting documentation of the

financial documents from Bresse that fromfor 1304 to 1307, suggest above-average summer temperatures certainly reached

above-average values. Additionally, the . Such a link existing between good wine harvestproduction and warm summer halfyear stands out ingrowing seasons exists for 1313 too. Accounts, as well, when account records 
[revised manuscript text omitted]

---

## Editor Decision (ED1)

Abstract. The cold/wet anomaly of the 1310s ("Dantean Anomaly") has attracted much attention from scholars, as it is commonly interpreted as a signal of the transition between the Medieval Climate Anomaly (MCA) and the Little Ice Age 10 (LIA). The huge variability  observed during this decade, similar to the high interannual variability observed in the 1340s, has been highlighted as a side effect of this rapid climatic transition. In this paper, we demonstrate that a multi-seasonal drought of almost two years occurred in the Mediterranean between 1302 and 1304, followed by a series of hot, dry summers north of the Alps from 1304 to 1306. We suggest that this outstanding dry anomaly, unique in the thirteenth and fourteenth centuries, together with cold anomalies of the 1310s and the 1340s, is part of the climatic shift from the MCA to the LIA. Our  reconstruction of the predominant weather patterns of the first decade of the fourteenth century—based on both documentary and proxy data—identifies multiple European precipitation seesaw events between 1302 and 1307, with similarities to the seesaw conditions which prevailed over continental Europe in 2018. It can be debated to what extent the 1302–1307 period can be compared to what is currently discussed regarding the influence of the phenomenon of Arctic amplification on the increasing frequency of persistent stable weather patterns that have occurred since the late 1980s. Additionally, this paper deals  with socioeconomic and cultural responses to drought risks in the Middle Ages, as outlined in contemporary sources, and provides evidence that there is a significant correlation between pronounced dry seasons and blazes that devastated cities.

1 Introduction & State of the Art

In recent decades, scholars of medieval studies have produced considerable research reconstructing the Little Ice Age (Pfister, Schwarz-Zanetti, Wegmann 1996) and appraising the impacts of cold events on premodern societies, but, apart from the notable exception of economic historians, few scholars have addressed the issue of droughts (Stone 2014). Almost two decades ago, Brown (2001) has highlighted the so-called "Dantean Anomaly" as a wet and cold anomaly lasting from 1315 to 1321, that led to famine over northwestern Europe (Jordan 1996). This climatic anomaly has been recently described more neutrally as "the 1310s event" (Slavin 2018). A distinctive "1300 event" has been found in proxy data, even around the Pacific rim (Nunn 2007). Historians have consistently focused on the cold, wet character of this decade, seemingly fascinated by continuous rains and their often detrimental impacts on food security. Much has been written, for example, about how excessive rain in 1315 and 1316 caused harvests to fail and ultimately resulted in a famine in northern Europe (Campbell 2016; Jordan 1996)

Commented [SG1]: Long complicated sentence structure, needs tightening up

Commented [SG2]: Not suitable term…better to use the term 'fires'

---

## Author Response (AR2)

GWZO, Reichsstraße 4–6, 04109 Leipzig

Prof. Stefan Grab
University of Whitswatersrand
Johannesburg, South Africa

Stefan.grab@wits.ac.za

Leipzig, 02.10.2020
Betreff:
Author's response

Dear Stefan,

please find attached our responses to the Referee comments and a marked-up version of our document. We haven't changed the figures, but apart from that, large parts of the manuscript structure have been revised according to RC2 and our own critical review of our first manuscript. And of course we integrated the feedback of RC1 (Christian Rohr), which required only minor formal changes. Furthermore, we had a thorough language copy-editing by a native speaker with a professional background in academic research in history. As linguistic quality was a key critique to our paper, well deserved, we hope to have changed it and present an article in much better quality now. We have furthermore, so we hope, corrected all the smaller formal mistakes that still were in the manuscript when we uploaded it fort he first time in mid-September.

Supplementary Information is attached as a zip file of one PDF and one Excel file.

I hope that's sufficient. Please come back to me if any questions are still open.

Thanks for all your efforts with our article,

Leibniz-Institut für
Geschichte und Kultur
des östlichen Europa
(GWZO) e.V.

Specks Hof (Eingang A)
Reichsstraße 4–6, 04109 Leipzig
Tel. +49 341 9735 587
Fax +49 341 9735 569
info@leibniz-gwzo.de
www.leibniz-gwzo.de

**Dr. Martin Bauch**
Projektleiter
Nachwuchsforschungsgruppe
"The Dantean Anomaly 1309-1321"
martin.bauch@leibniz-gwzo.de

Vereinsregister: Amtsgericht Leipzig, Registernummer: VR 2617
Finanzamt Leipzig II, St. Nr. 231/140/27308, Bankverbindung: Sparkasse Leipzig,
BIC: WELADE8LXXX, IBAN: DE21 8605 5592 1100 2775 24

[revised manuscript text omitted]